

# A malware detection method with function parameters encoding and function dependency modeling

Ronghao Hou[1], Dongjie Liu[1], Xiaobo Jin[2], Jian Weng[1] and Guanggang Geng[1]

[1] School for Cyberspace Security, Jinan University, Guangzhou, Guangdong, China
[2] Department of Electrical and Electronic Engineering, Xi'an Jiaotong-Liverpool University, Suzhou, Jiangsu, China

## ABSTRACT

As computers are widely used in people's work and daily lives, malware has become an increasing threat to network security. Although researchers have introduced traditional machine learning and deep learning methods to conduct extensive research on functions in malware detection, these methods have largely ignored the analysis of function parameters and functional dependencies. To address these limitations, we propose a new malware detection method. Specifically, we first design a parameter encoder to convert various types of function parameters into feature vectors, and then discretize various parameter features through clustering methods to enhance the representation of API encoding. Additionally, we design a deep neural network to capture functional dependencies, enabling the generation of robust semantic representations of function sequences. Experiments on a large-scale malware detection dataset demonstrate that our method outperforms other techniques, achieving 98.62% accuracy and a 98.40% F1-score. Furthermore, the results of ablation experiments show the important role of function parameters and functional dependencies in malware detection.

## INTRODUCTION

In recent years, with the widespread use of Internet-related applications in various aspects of people's work and lives, the diversity and quantity of malware have grown rapidly, continuously innovating in spreading methods. According to recent data, the number of malware and potentially unwanted applications (PUA) has risen year by year (*AV TEST, 2023*; *ENISA, 2023*). With the emergence of new organizations for advanced persistent threats (APT), the type, scale, and consequences of network attacks have increased significantly. Malware-related incidents surged in the first half of 2023 and show no signs of slowing down. Malware has become one of the most prevalent security threats. For example, when foundational services such as the domain name system (DNS) are targeted, malware can inflict widespread damage, including data exfiltration, data encryption for ransom, and system disruption (*Ni, 2019*; *Zheng, 2018*). If left unchecked, malware propagation poses severe risks to individuals, organizations, and society at large.

Corresponding author
Dongjie Liu, djliu@jnu.edu.cn

Therefore, malware detection, including the monitoring of emerging variants and malware families, is critical to system and network security.

Application programming interface (API) is a set of functions that define the interaction between software components, allowing different software systems to communicate and interact with each other, facilitating the interaction of data and functionality (*Ofoeda, Boateng & Effah, 2019*). For application software, the API provided by the operating system plays a crucial role in its development (*Microsoft, 2024*). Generally speaking, malware usually needs to directly or indirectly utilize the APIs provided by the operating system to achieve specific functions. For example, worms often use the file-system API to search, copy, and create files, as well as the network API to establish connections, transfer data, and communicate with other systems; Trojan viruses often use the remote code execution API to execute code on other systems. Ransomware calls the file-system API to access files.

Currently, many efforts have been made to apply artificial intelligence to analyze API call sequences for enhancing the accuracy of malware detection. Compared with full code analysis, API call sequence analysis is relatively lightweight, making real-time detection more efficient. In addition, API call sequence analysis focuses on interactions between programs and the operating system rather than specific code implementations, thus helping to detect both known and unknown malware variants. Importantly, APIs provide dynamic runtime information, allowing detection systems to observe and analyze malicious behavior in real time. Some researchers have used machine learning algorithms, such as K-nearest neighbor (KNN), naive Bayes (NB), decision tree (DT), support vector machine (SVM), and Random Forest (RF), to analyze API call sequences (*Singh & Singh, 2022*; *Amer et al., 2022*; *Amer & Zelinka, 2020*; *Sharma, 2022*; *Ndibanje et al., 2019*). Some researchers have focused on improving the accuracy of malware detection by employing deep learning methods for feature extraction (*Hemalatha et al., 2021*; *Shaukat, Luo & Varadharajan, 2023*; *Liu & Wang, 2019*; *Maniriho, Mahmood & Chowdhury, 2023*; *Zhang & Li, 2020*; *Li et al., 2022*; *Chen et al., 2022*). However, two factors still limit the effectiveness of API-based malware detection: the lack of function-parameter analysis and the absence of function dependency analysis.

First, existing research often focuses on API names, paying relatively little attention to API parameters. Both malware and benign software may call the operating-system–provided APIs to perform specific functions. For example, ransomware often calls the file-system API for file operations, whereas benign applications also use the file-system API to read files. Although malware and benign software may invoke the same API functions, malware typically uses more dangerous parameters, while benign software tends to use them less frequently. Therefore, analyzing API function parameters can effectively distinguish malware from benign software.

Secondly, it is crucial to effectively capture dependencies among API calls. Due to the complex relationships between API calls, API call sequences contain rich information. For example, APIs such as OpenProcess and CreateFile are used to obtain resource handles before calling CloseHandle. Additionally, CreateDirectory, CreateFile, and WriteFile are often used together to write a new file at a specific location. These relationships can reveal

software behavior. Therefore, effectively capturing dependencies among API calls can enhance malware detection.

Based on the above analysis, we propose a new malware detection method that leverages parameterized API call sequences. First, considering the characteristics of different API parameter types, we apply numerical normalization, term frequency–inverse document frequency (TF–IDF), and ordinal encoding to design encoders that extract features of each parameter type. Second, we employ a clustering-based discretization of parameters to produce a parameter-enhanced API call sequence, and we learn API semantics from many software samples *via* word embeddings to obtain semantic representations of function calls. Finally, we design a deep neural network combining temporal convolutional networks (TCN) and gated recurrent units (GRU) to model dependencies among API calls. Experimental results on a large-scale malware detection dataset (over 20,000 benign samples and 40,000 malware samples) demonstrate the effectiveness of our method. Furthermore, ablation studies show that modeling function parameters and function dependencies significantly improve the malware detection performance.

The contributions in this article are as follows:

- To address the lack of consideration of API runtime parameters, we propose a new encoding method for parameters of API: according to the characteristics of different types of API parameters, various feature engineering coding technologies are used to encode API parameters; Furthermore, clustering technology is used to discretize various parameter features, making it easier to generate semantic representations of API as deep learning networks.
- To address the problem of long API sequence dependencies, we design a deep neural network architecture to process the extracted features, which combines TCN and GRU, which can effectively capture the dependencies between API calls, thereby improving the performance of malware detection.
- We conduct extensive experiments on public datasets and verify the excellent performance of this method in malware detection. Through ablation studies, we explore the impact of modeling function API parameters and function dependencies on malware detection.

## RELATED WORK

In this section, the dynamic malware analysis is categorized into the executables based approach, API based approach, and API call sequence based approach.

### Executables based malware detection

Method based on executables typically utilizes machine learning or deep learning methods to extract patterns and features of malware from a large number of executables, achieving high-precision malware detection. For example, *Hemalatha et al. (2021)* propose an efficient malware detection system based on deep learning which uses the reweighted class-balanced loss function in the final classification layer of the DenseNet model to achieve significant performance improvements in classifying malware by handling

imbalanced data issues. *Shaukat, Luo & Varadharajan (2023)* first visualized portable executable files as images, then used a fine-tuned deep learning model to extract deep features from those images. Finally, they employed SVM for malware detection based on these deep features (*Shaukat, Luo & Varadharajan, 2023*). *Chaganti, Ravi & Pham (2022)* propose a deep learning-based bidirectional-gated recurrent unit-convolutional neural network (Bi-GRU-CNN) model for detecting and classifying Internet of Things (IoT) malware using ELF binary file byte sequences as input features. *Saleh, Li & Xu, (2018)* introduced a high-performance malware detection system that combines deep learning and feature selection methodologies to differentiate malware from benign traffic (*Alomari et al., 2023*). However, due to the rich information embedded in executables and their large file size, the time cost of detecting executable files is relatively high.

## API based malware detection

Methods based on API mostly focus more on whether the API is called and its frequency. They often overlook the impact of API parameters on API and usually employ manual analysis for the relationships between APIs. For example, *Singh & Singh (2022)* captured software API calls using the Cuckoo sandbox, selected multiple types of APIs, and used their invocation as features for optimizing machine learning algorithms to detect malware. *Amer et al. (2022)* proposed an Android malware detection technique based on API and permissions, collecting application features by obtaining the most frequently used API calls and permissions and utilizing machine learning algorithms for malware detection. *Amer & Zelinka (2020)* employed word2vec to extract contextual relationships between API sequences, cluster similar APIs, and ultimately detect malware based on a Markov chain. *Sharma (2022)* extracted important features for malware detection from API call sequences, invocation situations, and call frequencies obtained from the dynamic analysis of malicious and benign samples. They used the TF-IDF method to determine the importance of each feature in these feature sets and evaluated the feature effectiveness using machine learning algorithms such as decision trees, support vector machines, logistic regression, and k-nearest neighbors (*Sharma, 2022*). *Ndibanje et al. (2019)* calculated the called frequency of each API in each API sequence, represented each malware sample using this frequency vector, and then applied the KNN machine learning algorithm for feature extraction. While these works recognize the superior performance of API calls in malware detection, methods based on machine learning models often struggle to consider the internal dependencies within sequences adequately.

## API call sequence based malware detection

With the introduction of deep learning-related technologies, researchers have begun to explore malware detection from the perspective of API call sequences, utilizing deep learning models to model sequence data and uncover dependencies within the data. For example, *Liu & Wang (2019)* analyze malware and normal software samples using the sandbox software. They collect API calls, and remove duplicate API calls. Then they use vectorization techniques from natural language processing (NLP) to explore relationships between the APIs and obtain vectors representing API calls. Finally, they employ

**Table 1 Summary of literature review.**

| Topic | Related work | Problem |
|---|---|---|
| Executables based malware detection | *Hemalatha et al. (2021)*, *Shaukat, Luo & Varadharajan (2023)*, *Chaganti, Ravi & Pham (2022)*, *Alomari et al. (2023)* | High consumption of resources |
| API call based malware detection | *Singh & Singh (2022)*, *Amer et al. (2022)*, *Amer & Zelinka (2020)*, *Sharma (2022)*, *Ndibanje et al. (2019)* | Lack of attention to other features of API |
| API call sequence based malware detection | *Liu & Wang (2019)*, *Maniriho, Mahmood & Chowdhury (2023)*, *Zhang & Li (2020)*, *Kishore, Gond & Mohapatra (2024)*, *Zhou et al. (2024)*, *Li et al. (2022)*, *Chen et al. (2022)*, *Feng et al. (2024)*, *Zhao et al. (2023)* | Overly simplistic analysis of function parameters |

bi-directional long short-term memory (Bi-LSTM) for malware detection (*Liu & Wang, 2019*). *Maniriho, Mahmood & Chowdhury (2023)* utilize the tokenizer in the Keras framework for tokenizing and encoding APIs. They propose an automatic feature extraction approach based on convolutional neural networks (CNN) and bi-directional gate recurrent unit (BiGRU) deep learning architecture (*Maniriho, Mahmood & Chowdhury, 2023*). *Feng et al. (2024)* propose a novel documentation-augmented Windows malware detection framework to extract the information of official Windows API documentation and construct API graphs. *Zhou et al. (2024)* leverage a dynamic instrumentation tool to hook the target program to collect the API sequence and argument features. Then, it exploits a hierarchical attention network (HAN) model to analyze the API sequence features (*Zhou et al., 2024*). The above studies have overlooked the impact of API parameters on malware detection.

Recently, some researchers have started to explore malware detection from API sequences with parameters. For example, *Zhang, Qi & Wang (2020)* used a hash method to extract heterogeneous features from API names and run-time parameters. These features were further concatenated and input into a deep learning model that aggregates multiple gated CNN models and bidirectional long short-term memory (LSTM) (*Zhang, Qi & Wang, 2020*). *Kishore, Gond & Mohapatra (2024)* explored the implementation of machine learning in malware classification and analysis by enabling dynamic and adaptive threat recognition. *Li et al. (2022)* proposed a hybrid feature encoder for extracting semantic features from API names and parameters. Subsequently, an API call graph was derived from the API call sequence, transforming the relationships between API calls into structural information of the graph. Finally, a graph neural network was designed for malware detection (*Li et al., 2022*). *Chen et al. (2022)* introduced a classification method based on rules and clustering to evaluate the sensitivity of parameters to malicious behavior, obtaining a parameter-enhanced API call sequence. Based on this sequence, native embeddings and classification label embeddings were applied to API calls, connecting the two to represent the APIs. The embedded sequence was then input into a deep neural network to train a binary classifier for malware detection (*Chen et al., 2022*). *Zhao et al. (2023)* employ parameter-augmented semantic chains to improve the system's resilience to unknown parameters and design a deep learning model consisting of gated CNN, Bi-LSTM, and an attention mechanism to extract semantic features embedded within the API sequences and improve the overall detection accuracy.

In summary, the general description of the above literature is shown in the following Table 1. API call sequence based malware detection faces two main constraints currently: Existing research often only considers API names, with weak analysis of API parameters, and how to effectively capture relationships between APIs. Therefore, the API parameters are given more attention in this article, and a deep learning model that combines TCN and GRU is designed to capture the dependencies among APIs.

## METHODS

In this section, the proposed approach and the motivation behind it will be described in detail.

### System overview

The system architecture is shown in Fig. 1. The system has two stages: Function Parameters Encoding and Dependency Modeling.

During Function Parameters Encoding, as shown in Fig. 2, one API with parameters is first processed by Function Parameters Extraction, which encodes all parameters into vectors. Then it is clustered by the trained clustering model belonging to this API, and then the cluster to which these parameters belongs will be predicted. After Discretization of Real-valued Parameters, the parameters are replaced with their cluster labels. Finally, the new tokens are converted into vector representations *via* Word2Vec.

In the Function Parameters Extraction phase, API call sequences with parameters are input into the parameter encoder. It utilizes various techniques to encode API parameters of different types, ultimately transforming the raw parameters into feature vectors.

In the Discretization of Real-valued Parameters phase, the clustering-based approach is employed to generate rules for partitioning the parameter space. Parameters are partitioned based on the extracted features from encoding, and the partitioned results replace the original parameters. The real-valued parameters are discretized, resulting in the API call sequence with augmented parameters. Utilizing word embedding, context dependencies in the API call sequence are learned, yielding word embeddings that represent the APIs. Finally, the API call sequence is transformed into the sequence of API embeddings.

In the Dependency Modeling phase, the API embedding sequence is fed into our proposed deep neural network. A TCN layer extracts temporal features from the API call sequence. Subsequently, a GRU layer analyzes those features. Finally, the GRU's output is passed through a linear layer followed by a sigmoid activation function to classify whether the API call sequence corresponds to malware.

### Function parameters encoding

#### *Motivation of API run-time parameter*

As is well known, malware can directly or indirectly utilize the APIs provided by the operating system to achieve specific functions. Therefore, APIs are advantageous for malware detection. However, not only malware but also benign software may use these APIs. For instance, ransomware typically employs the file-system API for file operations,

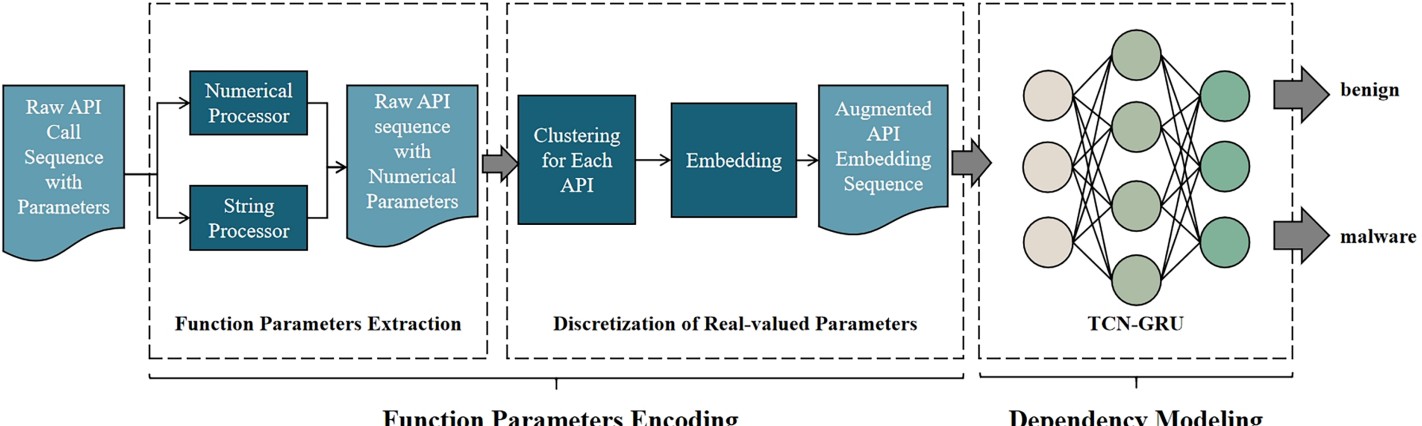

**Figure 1 Overview of our malware detection method.** The system mainly consists of two parts. (1) Function parameters encoding: to extract the feature of function parameters, discretize the feature of parameter, and embed API with the discretized parameters. (2) Dependency Modeling: to build a model that models the dependency of APIs, then train this model to classify.

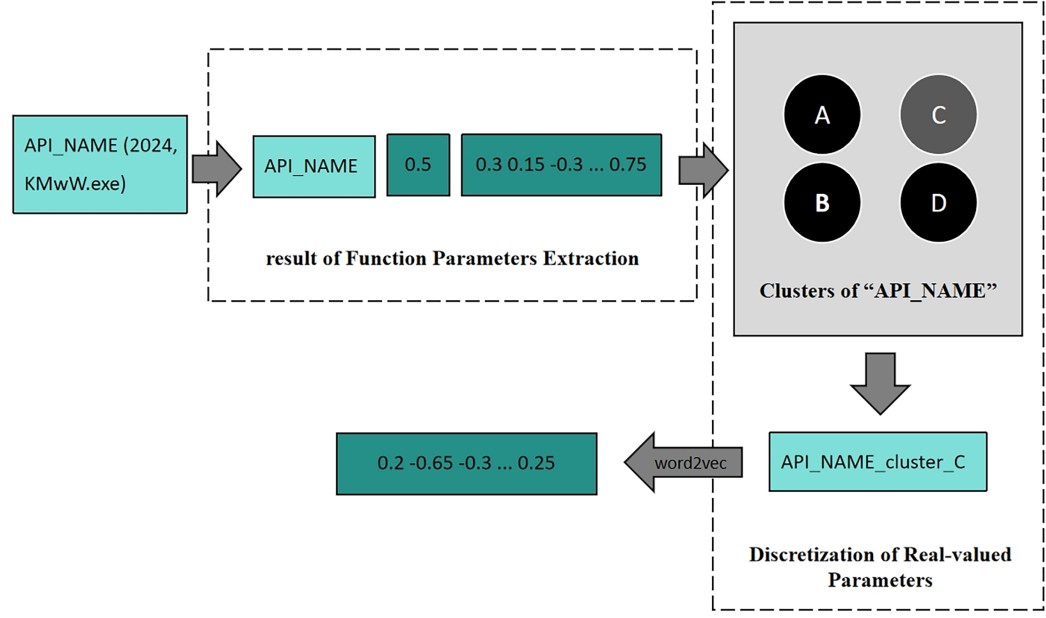

**Figure 2 An example of function parameters encoding.**

and benign software also uses the file-system API when reading external files (*Qbeitah & Aldwairi, 2018*; *Cozzi et al., 2018*; *Jamalpur et al., 2018*). Currently, most research does not take this into account and focuses only on raw API call sequences without additional context. While these studies have demonstrated some effectiveness, their performance has remained limited due to overlapping API usage between malware and benign software.

Fortunately, even though malware and benign software may use the same API, the former typically employs more dangerous parameters, which are often rarely used in the latter. Table 2 presents execution trace snippets of software samples in the sandbox.

**Table 2  Snippets of software traces.**

**(a) Malware**

NtCreateFile (“C:\Documents and Settings\admin\deadbeef\DEADBEEF”)

SetFileHiddenOrReadOnly (“C:\Documents and Settings\admin\deadbeef\DEADBEEF”, ReadOnly)

NtWriteFile (“C:\Documents and Settings\admin\deadbeef\DEADBEEF”)

WritePEFile (“C:\Documents and Settings\admin\deadbeef\DEADBEEF.exe”)

SetFileHiddenOrReadOnly (“C:\Documents and Settings\admin\deadbeef\DEADBEEF.exe”, Hidden)

Process32FirstW ()

CreateProcessInternalW (“C:\Documents and Settings\admin\deadbeef\DEADBEEF.exe”)

NtSetValueKey
 (“HKEY_LOCAL_MACHINE\SOFTWARE\Microsoft\Windows\CurrentVersion\Run”,
 “DEADBEEF.exe”)

…

**(b) Benign software**

NtCreateFile (“C:\User\lenovo\errorxxxxx.log”)

SetFileHiddenOrReadOnly (“C:\User\lenovo\errorxxxxx.log”, ReadOnly)

NtWriteFile (“C:\User\lenovo\errorxxxxx.log”)

…

…

…

NtSetValueKey (“HKEY_CURRENT_USER\SOFTWARE\Baidu\installDir”, “D:\Program
 Files\BaiduNetdisk”)

Typically, each line records information such as API names, process names, run-time parameters, return values, *etc.* For clarity, only API names and related parameters are listed here. In the execution snippet of malware, the malware calls NtCreateFile to create a new file. Then it writes malicious code into it using NtWriteFile and WritePEFile and sets the file as hidden using SetFileHiddenOrReadOnly. It finally modifies the registry using NtSetValueKey to enable malicious behavior to run automatically on the operating system startup. In the execution snippets of benign software, the benign software calls NtCreateFile, NtWriteFile, SetFileHiddenOrReadOnly to create a read-only error log recording run-time errors and uses NtSetValueKey to modify the registry, storing information such as the software's installation directory and version number. It can be observed that parameters such as 'ReadOnly', 'Hidden', and 'HKEY_LOCAL_MACHINE\SOFTWARE\Microsoft\Windows\CurrentVersion\Run' are sensitive and can aid in malware detection. Therefore, the behavior exhibited by API with different parameters is also viewed differently from a security perspective. This insight encourages us to utilize API parameters to enhance the effectiveness of malware detection.

### Function parameters extraction

As for run-time API parameters, their parameter space can be enormous, and the number and types of parameters vary significantly among different APIs. Using conventional embedding methods requires deeper neural networks. Alternatively, one can partition the

parameter space *via* rule-based schemes, but manually crafting rules for such diverse API parameters is time-consuming and labor-intensive.

If the first method is adopted, not only must the issue of how different parameters are encoded be addressed, but the challenge of aligning parameter features extracted in cases where the number and type of parameters vary across different APIs must also be faced. In contrast, the complexity of the problem is reduced by the second method, which uses rules to partition the parameter space into finite parts. Therefore, our approach to handling API parameters focuses on the second method, utilizing a clustering-based approach to automatically partition the parameter space. To ensure that the clustering results align with expectations, *i.e.*, categorizing API parameters into those associated with either secure or malicious behavior, the API parameters need to be encoded. They are transformed into features that represent their inherent security attributes. Directly using the raw string of API parameters is insufficient, as clusters formed by matching individual Unicode characters are unlikely to have any meaningful connection to malicious behavior.

Therefore, the parameter encoder is designed to reference feature engineering techniques for handling various types of parameters from the field of malicious behavior detection, such as phishing website detection, phishing email detection, and malicious code detection (*Yang, Zhao & Zeng, 2019*; *Unnithan et al., 2018*; *Saxe & Berlin, 2017*; *Saleh, Li & Xu, 2018*; *Yang et al., 2019*; *Zhang & Li, 2020*). The parameter encoder can encode security features for various types of API parameters. Parameters were categorized into two major types: numerical parameters and string parameters. Numerical parameters were further divided into two subtypes: integers and floating-point numbers. Complex string parameters, on the other hand, were further divided into seven subtypes, including IP address, file path, file name, URL link, registry, and others. For numerical parameters, different numerical parameters have different dimensions. To remove unit restrictions and facilitate the comparison of features with different units or magnitudes, numerical standardization is performed. For string parameters, they are semantically rich, but their types are complex and varied. The semantic information of a large number of string parameter types is hierarchical. Since parameters indicative of malicious behavior typically appear only in samples of malware, while parameters indicative of normal behavior appear in samples of both benign and malware, the TF-IDF algorithm (*Qaiser & Ali, 2018*) is suitable for this scenario. Therefore, this algorithm is used for feature encoding, and features such as the length of parameter strings, as well as the proportion of numerical characters, English characters, and special symbol characters, will be considered. More specifically, for parameters of an IP address, simply converting the four parts of an Internet Protocol version 4 (IPv4) address or the eight parts of an Internet Protocol version 6 (IPv6) address into numerical values is not sufficient. The features are added to determine whether it is an intranet or internet address. For parameters of the file path, the file path is divided, such as 'C:\Documents and Settings\admin\deadbeef,' into root directory 'C', first-level directory 'Documents and Settings', and second-level directory 'admin', encoding them separately to extract sensitive information from the path. For parameters of file name, both the file name and file extension will be extracted and encoded. For

---

**Algorithm 1   Function parameters encoding.**

**Require:** parameter_list (a list of input parameters)

**Ensure:** feature_vector (a feature vector representing the parameters)

 1: Initialize an empty list called `feature_vector`

 2: **for** each parameter in `parameter_list` **do**

 3:    Identify the type of parameter using regular expressions:

 4:    **if** parameter matches an integer pattern **then**

 5:        Normalize the integer

 6:        Append the normalized value to `feature_vector`

 7:    **esle if** parameter matches a floating-point pattern **then**

 8:        Normalize the floating-point number

 9:         Append the normalized value to `feature_vector`

10:    **else if** parameter matches an IP address pattern **then**

11:        Decimal form of the IP address

12:        Boolean value indicating if it is a private IP

13:        General string encoding

14:        Append all encoded parts to `feature_vector`

15:    **else if** parameter matches a file path pattern **then**

16:        Root directory encoding

17:        First-level directory encoding

18:        Second-level directory encoding

19:        General string encoding

20:        Append all encoded parts to `feature_vector`

21:    **else if** parameter matches a file name pattern **then**

22:        File name encoding

23:        File extension encoding

24:        General string encoding

25:        Append all encoded parts to `feature_vector`

26:    **else if** parameter matches a URL pattern **then**

27:        Top-level domain encoding

28:        Second-level domain encoding

29:        Subdomain encoding

30:        General string encoding

31:        Append all encoded parts to `feature_vector`

32:    **else if** parameter matches a registry pattern **then**

33:        Registry root key encoding

34:        Second-level registry key encoding

35:        Third-level registry key encoding

36:        General string encoding

37:        Append all encoded parts to `feature_vector`

38:    **else**

| Algorithm 1 (continued) |
|---|
| 39:   Append General string encoding to `feature_vector` |
| 40:  **end if** |
| 41: **end for** |

parameters of the uniform resource locator (URL) link, the domain name will be divided, and the top-level domain, second-level domain, and subdomain will be extracted for encoding. For parameters of a registry, similar to parameters of a file path, they will be divided into registry root keys, second-level registry keys, and third-level registry keys, and encoded separately. Details are shown in Algorithm 1. Where the general string encode method is TF-IDF algorithm, statistics parameter string length, and the proportion of numeric characters, English characters, and special symbol characters.

The original API call sequence from software samples is a list composed of strings, where each string represents an API call and its parameters in the form of 'API_NAME (Parameter_1, Parameter_2, …, Parameter_N)'. Specifically, when the function "CopyFileExW(KMwW.exe, C:\classified.doc.exe)" is passed into the encoder, it is handled as follows: First, all parameters are added to the parameter pool of the function named CopyFileExW for further processing. The pool stores the runtime parameters of CopyFileExW calls from the programs. Then the diverse types of parameters need to be identified. String matching techniques, specifically regular expressions, were employed to recognize and extract various types of parameters from the raw string of the API call sequence. In this example, "KMwW.exe" is recognized as a file, and "C:\classified.doc.exe" is also recognized as a file. Subsequently, the encoder encodes these parameters based on their types. For example, "KMwW.exe" is split into the file name and file extension, *i.e.*, "KMwW" and "exe". Each part is then encoded separately, with TF-IDF encoding applied to "KMwW.exe", "KMwW", and "exe". Additionally, statistics such as the string length of "KMwW.exe", the proportion of numeric characters, the proportion of alphabetic characters, and the proportion of special characters are also calculated. Finally, the word embeddings and parameter statistics are combined to form the feature vector for the API parameters.

### Discretization of real-valued parameters

After the Function Parameters Extraction, the API parameters are transformed into feature vectors. The next step involves using clustering methods to partition the parameter space based on these feature vectors related to malicious behavior. This process aims to cluster parameters associated with malicious behavior together and cluster those associated with benign behavior, achieving a finer-grained distinction for homonymic API calls with different API parameters.

Moreover, different APIs may have different types and quantities of parameters. For example, calling LoadLibrary needs to pass the parameter of the file name, while calling NtDeleteKey requires passing the parameter of the registry key. The API named LoadLibrary requires only one parameter, which is the dynamic link library (DLL), while

the API named NtSetValueKey needs to pass two parameters: the registry key and the value to be set. Therefore, all parameters of all APIs cannot simply be clustered together; instead, the API parameters of each API should be clustered separately.

The K-means algorithm is a widely used unsupervised learning method, primarily applied to clustering problems. The core idea is to partition the data into multiple independent clusters, minimizing the distance between data points within each cluster while maximizing the distance between clusters. It possesses advantages such as low time complexity and strong scalability (*Oti et al., 2021*). K-means++ is an improved version of the K-means algorithm, known for its ability to select better initial centroids, thereby enhancing the algorithm's convergence speed and reducing the risk of falling into local optima. Currently, it has been widely applied in various fields, including document clustering, customer classification, and anomaly detection, and has shown good performance. Therefore, considering our objectives and the characteristics of various clustering algorithms, we adopt the K-means++ algorithm as our clustering method.

In summary, in the Discretization of Real-valued Parameters, the K-means++ algorithm is used to cluster the safety feature vectors of all parameter lists for each API call. Through this, parameter lists of one API will be clustered to some clusters. After the completion of clustering, the original parameters will be replaced with the marking of the cluster, thus obtaining the parameter-enhanced API call sequence. Additionally, word2vec (*Di Gennaro, Buonanno & Palmieri, 2021*) is employed, treating API as words and software samples as sentences. This approach enables us to learn contextual dependencies in API call sequences and obtain word embeddings that represent the APIs with parameters.

## Dependency modeling

Previous research has often treated the problem of malware detection based on API call sequences as a sequence classification problem, typically employing deep learning models such as recurrent neural networks (RNNs) and LSTMs. Models of this kind (*Sherstinsky, 2020*; *Zargar, 2021*) determine each output based on both the current input and previous information. Therefore, they can handle sequence data, uncover temporal information in the data, and capture dependencies among sequence data. However, when dealing with long sequences, these models may encounter issues such as vanishing or exploding gradients as the network depth increases.

TCN is a deep neural network model based on one-dimensional convolution (*Bai, Kolter & Koltun, 2018*). Due to its ability to compute data from all time steps, as well as its powerful capability to model long-term dependencies with fewer parameters, TCN has been widely applied in areas such as speech recognition, motion detection, and time series classification. It is built on two mechanisms: causal convolution and dilated convolution.

Causal convolution is illustrated in Fig. 3. The value at position t in the output layer depends only on the values at position t and earlier in the input layer. Unlike traditional convolutional neural networks, causal convolution cannot see future data; it has a

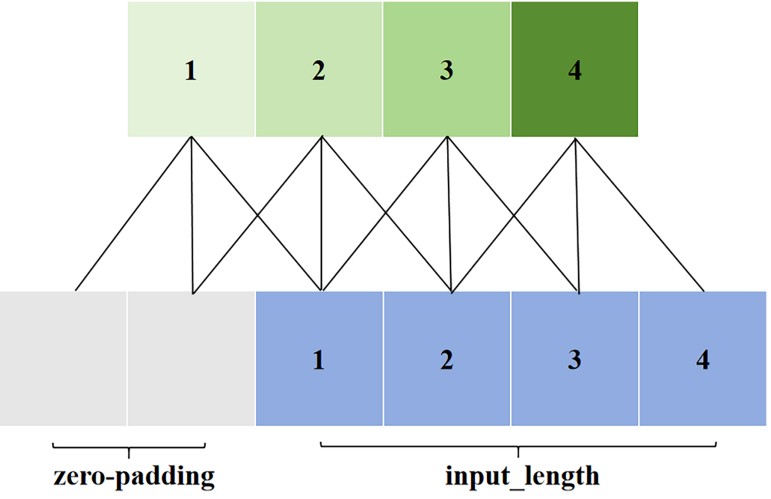

**Figure 3 Causal convolution in TCN.**

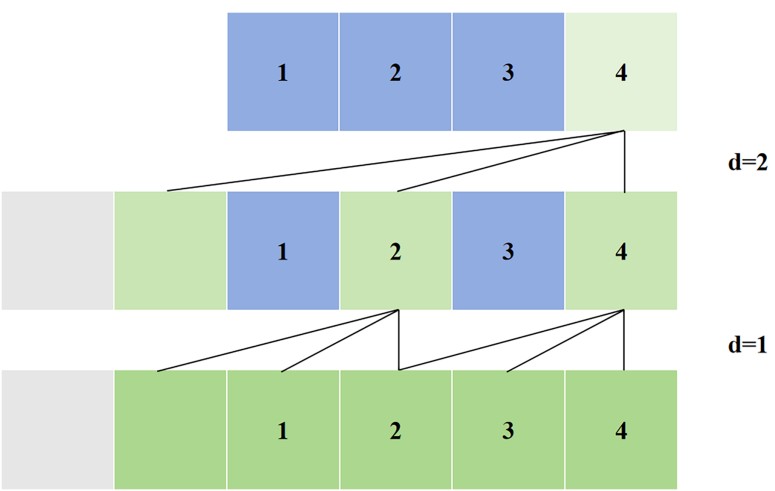

**Figure 4 Dilated convolution in TCN.**

unidirectional structure. In other words, there must be a cause before there is an effect, making it a strictly time-constrained model, hence the name causal convolution.

Simple causal convolution still has the problem of traditional CNN, that is, the length of modeling time is limited by the size of the convolution kernel, and it is difficult to obtain longer dependencies. The solution to this problem is dilated convolution, as shown in Fig. 4. Dilated convolution allows for interval sampling of the input during convolution, and the sampling rate is controlled by the dilated coefficient, *i.e.*, $d$ in the figure. $d = 1$ means that every point in the input process is sampled, and $d = 2$ means that every two points in the input process are sampled once as input. Generally speaking, the higher the level, the greater the value of $d$. Therefore, dilated convolution makes the size of the effective window increase exponentially with the number

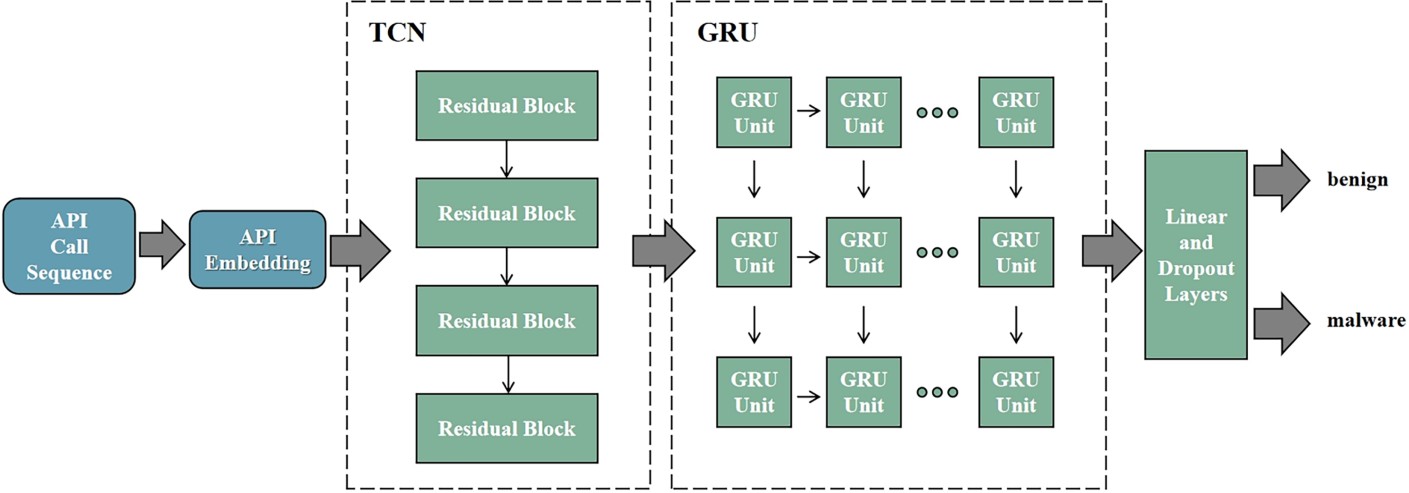

**Figure 5 TCN-GRU model architecture.** The TCN-GRU model consists of four parts. (1) Embedding: to embed API call sequence. (2) TCN: to explore the dependency relationships between APIs. (3) GRU: to extract temporal features of the API call sequence. (4) Linear: to classify where the API sequence is from.

of layers. In this way, the convolution network can use fewer layers to obtain a large range of receptive fields.

To ensure that the receptive field of the TCN covers the entire history, *i.e.*, the complete sequence, it is necessary to control the number of layers to achieve a wide receptive field.

For the given sequence length denoted as *l*, kernel size denoted as *k*, dilation base denoted as *b*, and number of layers denoted as *n*, the following inequality Eq. (1) needs to be satisfied to cover the complete history:

$$1 + (k-1) \cdot \frac{b^n - 1}{b - 1} \geq l. \tag{1}$$

Solving for *n* to obtain the minimum required number of layers as following Eq. (2):

$$n = \left\lceil \log_b \left( \frac{(l-1) \cdot (b-1)}{(k-1)} + 1 \right) \right\rceil. \tag{2}$$

The two mechanisms of causal convolution and expansion convolution are used to make the output of each moment in the TCN network only convoluted with the input at that moment and before. Therefore, the output of TCN maintains a temporal sequence. In other words, while TCN employs convolutional operations to extract features from the sequence, these features still possess temporal order. Consequently, TCN can be combined with neural networks such as RNN, LSTM, and GRU.

RNN is designed to handle sequential data and capture temporal dependencies within the input sequences. However, RNNs suffer from vanishing and exploding gradient problems, making it difficult to capture long-range dependencies. LSTM addresses this issue by introducing gating mechanisms (input gate, forget gate, and output gate) and memory cells, allowing them to retain important information over longer time horizons.

GRU, a simplified version of LSTM, contains only an update gate and a reset gate, resulting in fewer parameters and more efficient computation. GRU typically offers performance comparable to or even better than LSTM, especially in scenarios where model complexity and computational resources are limited. Therefore, by combining TCN with GRU, we construct the deep neural network shown in Fig. 5. The deep neural network is used to determine whether the input API call sequence is from malware. The model consists of the TCN layer, the GRU layer, and the Linear layer.

In summary, the model will take the API call sequence as input and determine whether the API call sequence originates from malware. The specific process is as follows: first, based on the trained API embedding from the previous step in API Embedding, construct the embedding layer to transform the input API call sequence into the API embedding sequence. Then, using the TCN layer, explore the dependency relationships between API calls, extracting temporal features of the API call sequence. Next, input the feature vectors output by the TCN layer into the GRU layer, overlaying bidirectional feature information. This further explores the intrinsic sequential correlations of the API call sequence in both forward and backward directions, extracting deeper temporal features. Finally, input the feature vectors extracted from the GRU layer into the fully connected layer. Through the sigmoid activation function, classify whether the input API call sequence is from malware.

## EXPERIMENT RESULTS

In this section, some details of the experiment and the experimental results are presented to evaluate our method.

### Dataset

An open dataset is used for evaluating our proposed method. The dataset is generated with a sandbox and is publicly available in GitHub, provided by the third-party (*kericwy1337, 2019*). They were also used as dataset in the malware detection track called DataCon2019 Big Data Security Analytics Competition. The dataset contains a total of 60,000 software samples, with 40,000 being malware and an additional 20,000 being benign. It is divided into two parts, in equal proportions based on label ratios. The malware detection model will be trained in the first part called the training set, and its performance will be tested in the second part, called the test set. Details of the dataset are shown in Table 3. The first part contains a total of 30,000 execution traces of Windows PE files, of which 10,000 are benign and the others are malicious. The second part also contains a total of 30,000 execution traces of Windows PE files, of which 20,000 are malware and the others are benign. Yet they are slightly different in the number of traced APIs, *i.e.*, 99 API in the first part and 101 API in the second part.

During the training, the five-fold cross-validation method is employed, where the training set is randomly divided into five equally sized subsets. Four subsets are used for training, and the remaining one is used for validation in each iteration. This process is repeated five times, and the average results are obtained. Simultaneously, the results on the

**Table 3 Detail of dataset.**

| Dataset | Benign | Malware | Involved API |
|---|---|---|---|
| Training set | 10,000 | 22,341 | 99 |
| Test set | 10,000 | 22,342 | 101 |

training set are referred to as training results, the results on the validation subset as validation results, and the results on the test set as test results.

## Experiment settings

In the proposed model, considering the median and mean of the API sequence lengths in the dataset, we have constrained the length of the API sequences to 1,000. Simultaneously, we set the size of the API embeddings to 50 based on the size of the *corpus* and the cost of unsupervised training of word2vec. Considering the complexity of the malware detection task and the length of the API call sequence, we also set the kernel size of the TCN model to three and the output channel of the TCN model to [100, 100, 100, 100, 100, 100, 100, 100, 100]. We set the hidden layer size of the GRU model to 64 to balance task complexity and training consumption. Finally, we set the dropout rate to 0.5 to avoid overfitting.

To prevent overfitting, we employ regularization techniques such as dropout. Additionally, we verify the model with a separate validation set and perform cross-validation to ensure the robustness of our results.

Additionally, the model applies Adam as the optimizer and supervises each input sequence with the label. To measure the loss of the training stage, the binary cross-entropy function is used as follows:

$$BCE = -(y \log(p) + (1-y) \log(1-p)), \tag{3}$$

where $y$ is binary label and $p$ is the probability of $y$.

Accuracy, Precision, Recall, and F1-score are used as the evaluation metrics of the proposed method:

$$Accuracy = \frac{|TP| + |TN|}{|TP| + |TN| + |FP| + |FN|}, \tag{4}$$

$$Precision = \frac{|TP|}{|TP| + |FP|}, \tag{5}$$

$$Recall = \frac{|TP|}{|TP| + |FN|}, \tag{6}$$

$$F1\text{-}score = 2 \times \frac{Precision \times Recall}{Presion + Recall}, \tag{7}$$

where TP represents the number of traces that are correctly predicted as malware, TN denotes the number of traces that are correctly classified as benign, FN denotes the number of traces that are malware but are incorrectly predicted as benign, and FP indicates the number of traces that are benign but are predicted as malware.

**Table 4 Validation results (%) of comparison with machine learning and deep learning model on malware detection.**

| Approach | F1-score | Precision | Recall | Accuracy |
|---|---|---|---|---|
| Naive Bayes | 65.96 | 67.69 | 70.90 | 67.08 |
| Decision tree | 89.47 | 91.86 | 87.78 | 91.51 |
| Logistic regression | 85.59 | 84.72 | 86.77 | 87.45 |
| SVM | 85.88 | 84.69 | 87.82 | 87.48 |
| TextCNN | 96.09 | 95.58 | 96.63 | 96.63 |
| RNN | 87.05 | 87.11 | 87.00 | 89.00 |
| Attention_BILSTM | 90.22 | 92.34 | 88.68 | 92.03 |
| Proposed method | 98.59 | 98.37 | 98.82 | 98.78 |

## Comparison with different model

At first, the proposed method is compared with four machine learning models (naive Bayes, decision tree, logistic regression, support vector machine) and three deep learning models (TextCNN, RNN, Attention_BILSTM) in terms of F1-score, Precision, Recall, and Accuracy. The specific approach includes training and validation on the training set and testing on the test set. The raw API sequences are input into all these models.

The experimental results, as shown in Tables 4 and 5, demonstrate that our proposed method outperforms the baseline models in the validation and test sets. For example, in the test results, our proposed model achieves F1-score, Precision, Recall, and Accuracy of 98.40%, 98.24%, 98.55%, and 98.62%, respectively. This represents an improvement of 8.64%, 6.22%, 10.40%, and 6.99% over the best baseline machine learning model and an improvement of 2.34%, 2.57%, 2.06%, and 2.01% over the best baseline deep learning model. The enhancement primarily comes from our proposed method, considering the significance of API run-time parameters in malware detection. In the model, the causal convolution and dilated convolution mechanisms of the TCN, maintain the temporal nature of the features extracted by the TCN. Additionally, GRU is deployed to comprehensively analyze the correlated features extracted by TCN, learning the deep temporal dependencies inherent in the data. Therefore, the proposed model achieves better performance, as validated by the experimental results.

## Comparison with state-of-the-art method

In order to evaluate our method, the proposed method is also compared with some methods presented by other researchers.

*Amer & Zelinka (2020)* employ the method of Markov chain to detect malware. Firstly, they employ Word2Vec to extract the contextual relationships between API in sequence and build the API similarity matrix. In accordance with the matrix, cluster similar APIs and replace API calls with cluster indexes. Then, the API sequence chain transition matrix is created by calculating the maximum transition sequence probability. Finally, the benign and malware probabilities of the tested sequence are calculated by traveling through the sequence *via* the transition matrices.

**Table 5 Test results (%) of comparison with machine learning and deep learning model on malware detection.**

| Approach | F1-score | Precision | Recall | Accuracy |
|---|---|---|---|---|
| Naive Bayes | 66.50 | 68.12 | 71.19 | 67.54 |
| Decision tree | 89.76 | 92.02 | 88.15 | 91.63 |
| Logistic regression | 86.03 | 85.21 | 87.11 | 87.73 |
| SVM | 86.16 | 85.05 | 87.97 | 87.64 |
| TextCNN | 96.06 | 95.67 | 96.49 | 96.61 |
| RNN | 87.13 | 87.30 | 86.98 | 89.05 |
| Attention_BILSTM | 90.45 | 92.45 | 88.98 | 92.16 |
| Proposed method | 98.40 | 98.24 | 98.55 | 98.62 |

*Zhang, Qi & Wang (2020)* utilize hashing methods to extract heterogeneous features from API names and run-time parameters. These features are then further concatenated and fed into a deep learning model that aggregates multiple gated CNN models and a bidirectional LSTM.

*Chen et al. (2022)* propose a rule and clustering-based classification approach to evaluate the sensitivity of parameters to malicious behavior and obtain the parameter-enhanced API call sequence. Based on this sequence, APIs are embedded natively and embedded with classification labels, connecting the two to characterize the API, and the API embedding sequence is fed into a deep neural network to train a classifier to detect malware.

*Maniriho, Mahmood & Chowdhury (2023)* propose API-MalDetect, a deep learning-based framework for automated malware detection. The framework employs an encoder to process API calls, combined with a hybrid feature extraction approach that integrates CNN and GRU to extract relevant patterns from sequences of API calls.

*Kishore, Gond & Mohapatra (2024)* proposed a method of malware detection based on machine learning. Several models are used for feature learning and classification. Finally, their experiment proves that XGBoost accurately detects malicious samples.

*Zhou et al. (2024)* propose a hybrid model, which combines the HAN and MLP models. It leverages a dynamic instrumentation tool to hook the target program to collect the API sequence and argument features. Then, it exploits a Hierarchical Attention Network model to analyze the API sequence features. Finally, it applies a multi-layer perceptron (MLP) model to analyze features.

Considering that the samples in the test set are almost entirely different from those in the training set, with potential new malicious samples in the test set, it can be used to evaluate the generalization ability of our model. Therefore, 5-fold cross-validation is used to train the model on the training set and then test the model on the test set. We primarily considered four metrics, including accuracy, precision, recall, and F1-score.

As shown in Table 6, the performance of the proposed model surpasses other models in terms of accuracy, precision, recall, and F1-score. Compared to machine learning-based

**Table 6 Comparison with state-of-the-art research on malware detection.**

| Method | Parameter | Validation results (%) | | | | Test results (%) | | | |
|---|---|---|---|---|---|---|---|---|---|
| | | F1-score | Precision | Recall | Accuracy | F1-score | Precision | Recall | Accuracy |
| Markov Chain (*Amer & Zelinka, 2020*) | No | 85.21 | 82.76 | 87.95 | 85.64 | 71.90 | 60.14 | 89.32 | 66.31 |
| Multiple Gated CNN and Bi-LSTM (*Zhang & Li, 2020*) | Yes | 96.04 | 95.78 | 96.32 | 96.85 | 87.96 | 94.94 | 82.62 | 89.23 |
| CNN and Bi-GRU (*Maniriho, Mahmood & Chowdhury, 2023*) | No | 92.65 | 92.58 | 92.71 | 93.70 | 92.67 | 92.46 | 92.89 | 93.71 |
| XGBoost (*Kishore, Gond & Mohapatra 2024*) | Yes | 94.07 | 94.83 | 93.40 | 95.07 | 94.18 | 94.89 | 93.55 | 95.10 |
| HAN (*Zhou et al., 2024*) | no | 96.10 | 96.27 | 95.95 | 96.67 | 96.51 | 96.45 | 96.57 | 97.01 |
| TextCNN and Bi-LSTM (*Chen et al., 2022*) | Yes | 96.23 | 95.75 | 96.71 | 97.03 | 96.54 | 96.26 | 96.91 | 97.33 |
| Our Proposed Method | Yes | 98.59 | 98.37 | 98.82 | 98.78 | 98.40 | 98.24 | 98.55 | 98.62 |

approaches proposed by *Amer & Zelinka (2020)*, our deep learning-based model achieves superior performance in malware detection. For instance, in the test result, our proposed method achieves 98.40% in F1-score, while the method proposed by *Amer & Zelinka (2020)* achieves 71.90%. These results indicate that deep learning models can capture deeper dependencies in the data, leading to better performance. Furthermore, the results suggest that combining API calls and associated run-time parameters significantly improves performance compared to models that do not consider parameters.

Compared to the deep learning models proposed by *Zhang & Li (2020)*, *Maniriho, Mahmood & Chowdhury (2023)*, *Kishore, Gond & Mohapatra (2024)*, *Zhou et al. (2024)*, and *Chen et al. (2022)*, our proposed model also demonstrates significant performance improvement. For instance, in the test results, our model outperforms the models of *Zhang & Li (2020)*, *Maniriho, Mahmood & Chowdhury (2023)*, *Kishore, Gond & Mohapatra (2024)*, *Zhou et al. (2024)*, and *Chen et al. (2022)* by 10.44%, 3.3%, 15.93%, 9.4%, and 5.73%, 5.78%, 5.66%, 4.91%, and 4.22%, 3.35%, 5%, 3.52%, and 1.83%, 1.71%, 1.98%, 1.51%, and 1.86%, 1.98%, 1.63%, 1.28% in terms of F1-score, precision, recall, and accuracy, respectively. Our proposed model not only takes into account the impact of API parameters on malware detection but also conducts a more in-depth and detailed treatment of API parameters. Experimental results validate the effectiveness of these treatments, leading to superior performance in the experiments compared to others.

# DISCUSSION

In this section, the effects of the improvements made in the proposed approach are discussed through the ablation study first. Then, the impact of varying the number of clusters on model performance is discussed.

Additionally, the receiver operating characteristic (ROC) curve is plotted to evaluate these effects. The ROC curve is a graphical representation commonly used in binary classification to assess the performance of a model across different decision thresholds. Plotted with the true positive rate (TPR) against the false positive rate (FPR), the ROC curve illustrates the trade-off between correctly identifying positive instances and incorrectly classifying negative instances. The area under the ROC curve (AUC) serves as a

**Table 7 Ablation study on TextCNN with Raw API, TextCNN with augmented API, TCN-GRU with Raw API and complete model.**

| Performance | Validation results (%) | | | | Test results (%) | | | |
|---|---|---|---|---|---|---|---|---|
| | TextCNN with raw API | TextCNN with augmented API | TCN-GRU with raw API | Complete model | TextCNN with raw API | TextCNN with augmented API | TCN-GRU with raw API | Complete model |
| F1-score | 90.22 | 96.49 | 96.52 | 98.59 | 90.45 | 96.04 | 96.37 | 98.40 |
| Precision | 92.34 | 96.21 | 95.59 | 98.37 | 92.45 | 95.67 | 95.49 | 98.24 |
| Recall | 88.68 | 96.79 | 97.65 | 98.81 | 88.98 | 96.43 | 97.44 | 98.55 |
| Accuracy | 92.03 | 96.95 | 97.00 | 98.78 | 92.16 | 96.58 | 96.83 | 98.62 |

comprehensive metric, quantifying the model's ability to discriminate between positive and negative instances. A higher AUC signifies better overall performance, and the curve provides valuable insights for selecting an appropriate threshold based on the specific requirements of the task at hand. Among them, the equations for TPR and FPR are as follows:

$$TPR = \frac{TP}{TP + FN},$$ (8)

$$FPR = \frac{FP}{TN + FP},$$ (9)

where true positive (TP), false negative (FN), false positive (FP) and true negative (TN) are the same as before.

## Ablation study

As mentioned above, our proposed method mainly consists of two improvements: first, the enhancement of API call sequences through encoding and clustering of run-time parameters; second, the introduction of a new deep neural network combining TCN and GRU. To assess the benefits of these two improvements, the performance differences before and after these enhancements will be compared. Furthermore, the TextCNN model performed the best among the commonly used models in the experiment of the previous section. So this involves evaluating the TextCNN model with raw API embeddings (TextCNN with raw API), the TextCNN model with parameter-augmented API embedding (TextCNN with augmented API), the TCN-GRU model with raw API embeddings (TCN-GRU with raw API), and the complete model. The experimental results are presented in Table 7.

Whether on the validation set or the test set, as expected, our proposed method improved all four metrics. Specifically, as the model progressed from the baseline model with raw API embeddings to the baseline model with parameter-augmented API embedding, then to the TCN-GRU model with raw API embeddings, and finally to the complete model, the F1-score on the test results increased from 90.45% to 96.04% and 96.37%, finally reaching 98.40%. These results demonstrate that, on one hand, our clustering effectively utilizes run-time parameters, providing a finer partition for API with different run-time parameters, assisting the deep neural network in learning more complex dependencies inherent in API call sequences. On the other hand, it proves that our

proposed combined TCN and GRU deep neural network model overcomes the issue of long sequence gradient vanishing present in other deep learning models. It effectively captures the implicit dependencies in sequence data, providing more efficient modeling of API call sequences.

Parameter encoding allows the APIs with the same name, which would normally be treated as identical symbols, into multiple distinct representations. Although different software samples may call the same APIs, the varying parameters assigned during execution lead to different behaviors. The method is good at capturing these behavioral differences caused by parameter variations, resulting in improved detection performance.

By using TCN-GRU to model the dependencies within API sequences, our approach benefits from the causal and dilated convolutions, which allow it to capture long-range contextual dependencies. This enhances its ability to resist obfuscation attacks that introduce the meaningless APIs. Moreover, the method effectively extracts deep features from API sequence dependencies, enabling it to achieve superior detection results.

Figure 6 illustrates the ROC curves of different models on the test set. As expected, the AUC values consistently increase as the model progresses from the model progressed from the baseline model with raw API embeddings to the baseline model with parameter augmented API embedding, then to the TCN-GRU model with raw API embeddings, and finally to the complete model, starting from 0.9794, reaching 0.9822 and 0.9891, and ultimately reaching 0.9985.

In summary, the two proposed improvements in this article effectively enhance the capability of malware detection. Moreover, for models trained on different datasets, they exhibit good robustness on the test dataset, demonstrating their generalization ability and proving their effectiveness in practical applications.

The confusion matrix as Fig. 7 highlights, that the model demonstrates excellent performance in detecting malware, as evidenced by its high precision and recall. The low number of false positives and false negatives ensures both reliability and robustness in malware classification. Our method exhibits strong detection capabilities with high precision, recall, and accuracy, making it well-suited for malware classification.

## Effect of various settings

The performance of the proposed model is mainly influenced by the method of clustering and the number of clusters in API parameter clustering. In this subsection, the impact of factors will be examined by configuring the proposed method with different settings.

In our approach, a clustering-based approach is used to achieve the discretization of API parameters, where we need to aggregate different combinations of parameters of the same API into different clusters to distinguish between normal and malicious behaviors of the same API. In this case, the clustering method can be a key factor affecting the effectiveness of malware detection. Therefore, the experiments test the impact of different clustering methods on the performance, mainly selected division-based clustering method (K-means), hierarchy-based clustering method (BIRCH; *Zhang, Ramakrishnan & Livny, 1996*), and short text clustering method (GSDMM; *Yin & Wang, 2014*). Table 8 compares the performance under different clustering methods. In the test results, the K-means

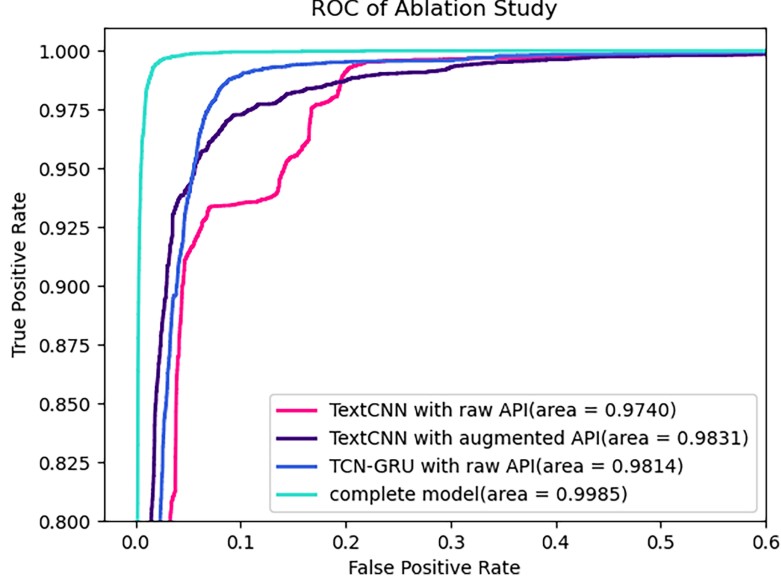

**Figure 6  ROC of ablation study on test set.**

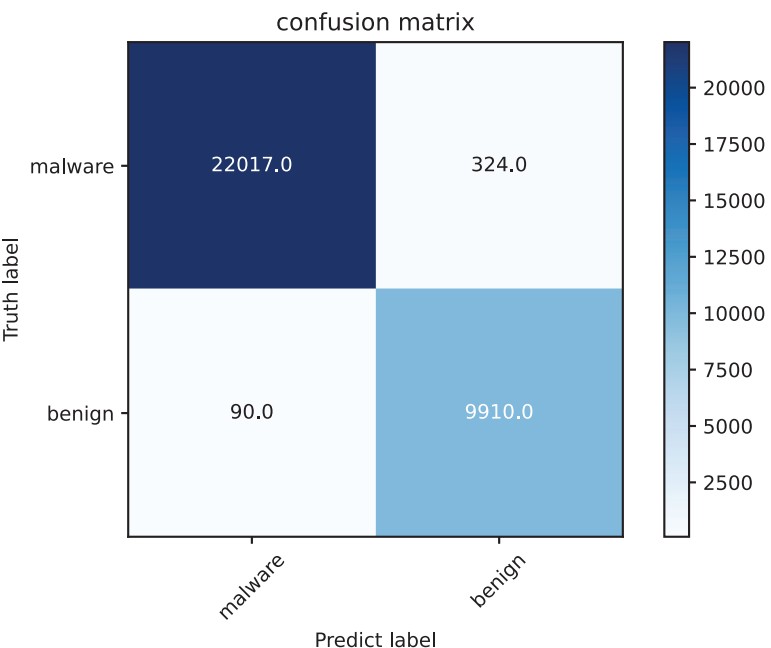

**Figure 7  Confusion matrix of our method on test set.**

achieves more outstanding results, improving the F1-score compared to the BIRCH and GSDMM methods by 0.3%, 5.6%. This also validates our initial idea that for the final task of binary classification, such as malware detection, where the encoder also encodes the features of the input parameters for clustering into a more continuous form, the data may

be relatively linear. In this case, the data may be relatively linearly divisible, which makes it easy to cluster into multiple, relatively centralized clusters under K-means.

Obviously, the number of clusters (denoted by K) is a very important setting that determines the granularity at which API calls with parameters are partitioned, and hence the granularity of the security semantics. In general, a smaller K will allow more different parameters to share the same security semantics, while a larger K enables a finer-grained characterization of the API's level of security. Table 9, Figs. 8, and 9 compare the performance for different cluster numbers. For instance, in the test results, the model with K = 5 achieved the peak F1-score, precision, recall, and accuracy, with a slight decline in performance for other values of K. Additionally, in the ROC curve, it can be observed that the model's AUC reaches the high of 0.9985 when K is 5 and gradually decreases as K increases. In examining the precision-recall curve (PR Curve), it is evident that the optimal curve is achieved when K is set to 5. This suggests that a clustering parameter of K = 5 yields the best performance in terms of precision and recall. This can be understood as, when the number of clustering labels is too small, it is challenging to effectively differentiate the security semantics of parameters, and when the number is too large, it may lead to the blurring of security semantics.

## Limitation

The main limitation of our current approach lies in its reliance on sandbox software for the dynamic analysis of software samples. This method requires the sandbox to extract runtime API sequences and their parameters from the samples. The analysis time required by the sandbox becomes a bottleneck in the overall malware detection process, especially when dealing with large volumes of software samples.

Additionally, the method introduces an added layer of complexity by incorporating API parameter analysis and modeling, which increases computational overhead. A promising direction for future research could involve embedding the approach directly into sandbox software. By integrating malware detection within the dynamic analysis phase, computational complexity will be decreased. The advantage of this integration would be the ability to identify malware during the dynamic analysis process itself, rather than requiring a complete analysis of the sample followed by a separate malware detection pipeline.

## Future work

Currently, our approach is applied in the field of malware detection. However, if an appropriate malware classification dataset were available, there is no doubt that our method could be extended to malware classification. Different types of malware exploit distinct techniques. For example, Adware tends to download other security tools, ransomware encrypts files on the hard drive, and worms propagate across networks to infect other machines. Our method models API and their parameters, enabling it to effectively identify these exploitation behaviors. By leveraging the TCN-GRU network, the corresponding dependency features can be extracted, demonstrating significant potential for application in this area.

**Table 8 Effect of varying method of clustering.**

| Method | Validation results (%) | | | | Test results (%) | | | |
|---|---|---|---|---|---|---|---|---|
| | F1-score | Precision | Recall | Accuracy | F1-score | Precision | Recall | Accuracy |
| GSDMM | 93.37 | 92.21 | 95.00 | 94.09 | 92.80 | 91.57 | 94.61 | 93.61 |
| BIRCH | 98.47 | 98.03 | 98.94 | 98.67 | 98.25 | 97.82 | 98.71 | 98.49 |
| K-means | 98.59 | 98.37 | 98.82 | 98.78 | 98.40 | 98.24 | 98.55 | 98.62 |

**Table 9 Effect of varying numbers of clusters.**

| K | Validation results (%) | | | | Test results (%) | | | |
|---|---|---|---|---|---|---|---|---|
| | F1-score | Precision | Recall | Accuracy | F1-score | Precision | Recall | Accuracy |
| 3 | 97.42 | 96.86 | 98.02 | 97.79 | 97.47 | 96.95 | 98.04 | 97.81 |
| 4 | 96.97 | 96.14 | 97.94 | 97.38 | 97.04 | 96.32 | 97.89 | 97.43 |
| 5 | 98.59 | 98.37 | 98.82 | 98.78 | 98.40 | 98.24 | 98.55 | 98.62 |
| 6 | 97.79 | 97.19 | 98.47 | 98.10 | 97.74 | 97.20 | 98.33 | 98.04 |

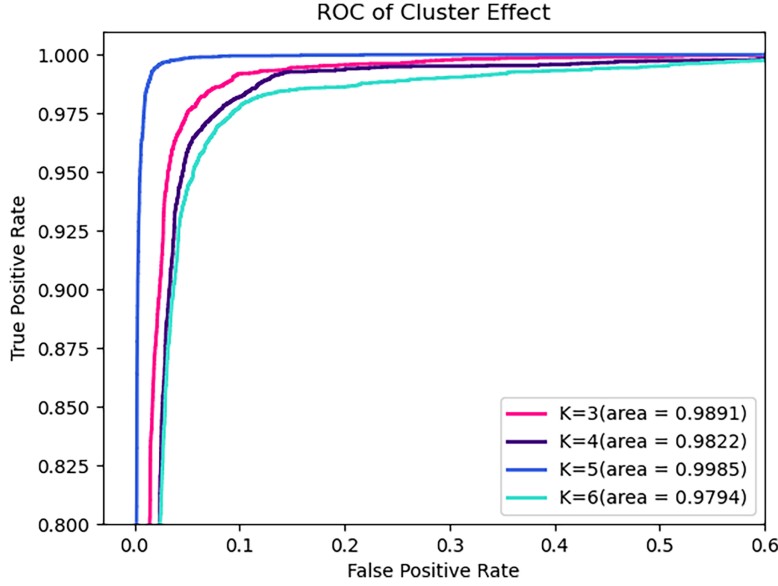

**Figure 8 ROC of varying numbers of clusters on test set.**

Furthermore, in addressing zero-day attacks, our approach differs from traditional methods that focus on defending against signature-based malware features. By analyzing API call sequences made by software, the method is better equipped to detect previously unseen zero-day attacks. Regardless of which new vulnerability an attacker exploits, the ultimate goal remains to acquire sensitive system information or encrypt user files—behaviors that our method can detect. Therefore, zero-day attack detection is within range.

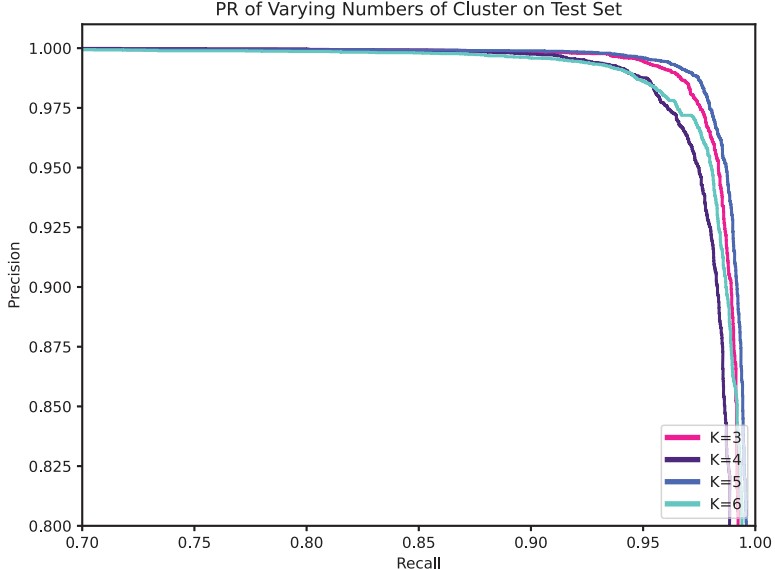

**Figure 9** **PR of varying numbers of clusters on test set.**

## CONCLUSIONS

In this article, we propose a malware detection method based on function parameter encoding and functional dependency modeling. Specifically, we employ various feature engineering techniques to extract security features for different types of parameters. Subsequently, the K-means algorithm is used to discretize various types of function participants to facilitate word2vec encoding the parameters of the function. Meanwhile, we design a deep neural network combining TCN and GRU to capture the dependencies between API calls. Experimental results show that our proposed method performs well in detecting malware, surpassing other classic methods. Furthermore, ablation studies demonstrate that our functional parameter encoding and functional relationship modeling play a crucial role in improving malware detection performance. In future work, we will seek better methods for encoding function parameters and modeling functional dependencies.

## ACKNOWLEDGEMENTS

This work would like to thank OpenAI for providing ChatGPT, which was used only for the translation of the sections of this manuscript from the author's native language to English.

### Funding

This work was supported by the National Key Research and Development Program of China through project 2023YFB3105700. Qing Lan Project in Jiangsu universities, XJTLU RDF-22-01-565020. Guanggang Geng is supported by the Pearl River Talents Plan. The

funders had no role in study design, data collection and analysis, decision to publish, or preparation of the manuscript.

## Grant Disclosures
The following grant information was disclosed by the authors:
National Key Research and Development Program of China: 2023YFB3105700.
Qing Lan Project in Jiangsu universities: XJTLU RDF-22-01-565020.
Pearl River Talents Plan.

## Competing Interests
The authors declare that they have no competing interests.

## Author Contributions
- Ronghao Hou conceived and designed the experiments, performed the experiments, analyzed the data, performed the computation work, prepared figures and/or tables, and approved the final draft.
- Dongjie Liu performed the experiments, analyzed the data, performed the computation work, prepared figures and/or tables, and approved the final draft.
- Xiaobo Jin conceived and designed the experiments, analyzed the data, performed the computation work, authored or reviewed drafts of the article, and approved the final draft.
- Jian Weng analyzed the data, prepared figures and/or tables, authored or reviewed drafts of the article, and approved the final draft.
- Guanggang Geng conceived and designed the experiments, analyzed the data, authored or reviewed drafts of the article, and approved the final draft.

## Data Availability
The data is available at Zenodo:

- kericwy1337, & brooomisname. (2025). getsword/Datacon2019-Malicious-Code-DataSet-Stage1: Datacon2019-Malicious-Code-DataSet-Stage1 (v1.2). Zenodo. https://doi.org/10.5281/zenodo.15285406

- kericwy1337. (2025). getsword/Datacon2019-Malicious-Code-DataSet-Stage2: Datacon2019-Malicious-Code-DataSet-Stage2 (v1.0). Zenodo. https://doi.org/10.5281/zenodo.14925179.

The code is available at Zenodo: brooomisname. (2025). getsword/malware_detection_with_API: malware_detection_with_API (v1.1). Zenodo. https://doi.org/10.5281/zenodo.14935798.

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
