# Peer review of "A malware detection method with function parameters encoding and function dependency modeling"

_PeerJ Computer Science, doi:10.7717/peerj-cs.2946_

## Round 0.1 · original submission · Major Revisions

Dear Authors,

Thank you for submitting your manuscript. Feedback from the reviewers is now available. It is not recommended that your article be published in its current format. However, we strongly recommend that you address the issues raised by the reviewers and resubmit your paper after making the necessary changes. Before submitting the paper following should also be addressed:

1. Too much first-person pronouns should be avoided. For example there are three in this sentence "We use Accuracy, Precision, Recall, F1-score as our evaluation metrics of our method."
2. Please pay special attention to the usage of abbreviations. Spell out the full term at its first mention, indicate its abbreviation in parenthesis and use the abbreviation from then on.
3. Many of the equations are part of the related sentences. Attention is needed for correct sentence formation.

Warm regards,

Reviewer 1 ·

Basic reporting

1) Some sections, especially the methodology, could benefit from simplified language to improve readability. Breaking down complex steps or adding pseudocode for intricate processes (such as parameter encoding or the TCN-GRU workflow) could help readers replicate the study more easily.

2) Reviewing recent hybrid and visualization-based approaches or methods that combine deep learning with traditional techniques could provide a broader context and highlight where this method stands among the latest developments in the field.

Experimental design

1) Discussing possible extensions of the method, such as adapting it to various malware types, handling zero-day attacks, or integrating with other security data sources, could open new directions for the research and highlight its potential impact.
2) Some figures, such as the model architecture, could use additional annotations or descriptive captions to clarify each component’s role. Improved visuals can help readers understand the workflow and structure of the model at a glance, making the paper more engaging and accessible.

Validity of the findings

1) While the introduction effectively presents the gaps in existing malware detection techniques, it could be enhanced by clearly linking these gaps to the benefits of the proposed method. For example, more explicit statements about why parameter encoding and dependency modeling provide better detection outcomes could underscore the study’s importance.
2) Adding a discussion on the practical aspects of implementing the model in real-world settings, such as its computational demands, runtime, or limitations with unseen or evolving malware patterns, would provide a more balanced perspective. This can also make the research more appealing to industry professionals.

Reviewer 2 ·

Basic reporting

Authors should do the extensive literature review on their specific problem statement.

Experimental design

Experimental methodology is not clearly explained and technically poor.

Validity of the findings

Dataset details have not been given. Results and comparisons should be well explained.

Additional comments

1. Extensive literature review with recent studies has to be done. Authors are suggested to include a table of summary of literature review (Related works).

2. Authors stated that, “a parameter feature encoder was designed for the various types of parameters”. The detailed process of parameter feature encoder along with algorithm with an example is required.

3. Figure1 is a high-level diagram. Authors are suggesting to include some more low-level description along with diagrams are needed.

4. Function parameter extraction and how the feature vector has been generated should be well explained.

5. The procedure of Discretization of Real-valued Parameters also not explained clearly.

6. Authors should include more details of their used dataset with clear citation.

7. Authors should also include the TCN-GRU model summary.

8. Some more graphs for model accuracy and confusion matrix are required.

---

## Round 0.2 · Major Revisions

Dear Authors,

The reviews for your revised manuscript are included at the bottom of this letter. We ask that you make changes to your manuscript based on those comments.

Best wishes,

Reviewer 2 ·

Basic reporting

Some typos and grammatical errors are observed in the manuscript. Authors are suggested to avoid these errors.

Authors are suggested to include some more recent works in their study,

Experimental design

The methodology section should be improved with clear description and implementation details.

Validity of the findings

Authors have compared thier results with the state-of-the-are works but still Results should be compared with some more recent works.

Additional comments

Review Comments:
1. Authors are suggested to add some more recent related studies in related work section.
2. The proposed methodology should be explained clearly. (Authors have stated, “According to the characteristics of different types of API parameters, various feature engineering coding technologies are used to encode API parameters; “ But not explained with any example. The entire methodology is like general statements. How the methodology is implemented should be well explained.)

Reviewer 3 ·

Basic reporting

The paper is well-organized with a clear introduction, related work section, methodology, and results discussion.

It is written in clear and unambiguous English.

The problems of the studies in the literature are stated, but no information is given about how the proposed model contributes to the existing problems or how it will improve the existing solutions. This clarity will provide a better understanding of the contributions of the paper.

Experimental design

In the "Experiment Settings" section, it would be useful to provide a more detailed explanation of how parameter values ​​were selected. In particular, more information should be provided on what criteria certain parameters were selected and how these choices affected the experimental results.

Comparisons should be made with state-of-the-art studies. Hybrid methods should be investigated and their performances should be evaluated.

While K-means++ is a solid choice, exploring other clustering algorithms such as DBSCAN or hierarchical clustering could provide deeper insights into the parameter space, especially in cases where the data is not linearly separable.

Validity of the findings

The model achieves impressively high accuracy on both the training and test sets, but there is no discussion about overfitting or how it might have been mitigated. A brief analysis of potential overfitting or overtraining would be beneficial.

---

## Round 0.3 · Major Revisions

Dear Authors,

Please clearly address the concerns and criticisms of Reviewer 2 and resubmit your paper once you have updated it.

Best wishes,

Reviewer 2 ·

Basic reporting

Basic reporting is poor. Sentences are not clear and no pfofessional english is used throut the manuscript.

Experimental design

Experimental design and methodology section is not properly addressed even in revised manuscript.

Validity of the findings

Results have not compared with very recent works even in revised form.

Additional comments

Overall clarity, methodology and implementation are not clear. These sections should be well addressed.

Reviewer 3 ·

Basic reporting

Clear and comprehensive explanations are provided on how the model contributes to studies or problems in the literature. The innovative aspect of the article is highlighted with concrete contributions such as coding of API parameters, processing of long API dependencies and demonstrating the effect of the parameters of the model with experiments.

Experimental design

The criteria for selecting parameter values ​​are explained. The reasons why the parameters are selected, how they are determined, and their effects on the experimental results are explained and the necessary technical details are presented.

The model was compared in detail with the state-of-the-art and hybrid methods using precision, recall, F1-score and accuracy metrics. The performance increase was demonstrated with concrete data. By adding hybrid methods, the success of the study was put forth in a comprehensive manner.

By comparing the results with alternative clustering methods, it is more clearly shown why the K-means method was chosen, along with its success in working with linear data.

Validity of the findings

Effective measures have been taken to prevent overfitting by using dropout, validation set and cross-validation. However, a more detailed analysis of this process could have been presented.

---

## Round 0.4 · Major Revisions

Dear Authors,

Please clearly and appropriately address the concerns and criticisms of Reviewer 2 and resubmit your paper once you have updated it.

Best wishes,

**Language Note:** The review process has identified that the English language must be improved. PeerJ can provide language editing services - please contact us at [email protected] for pricing (be sure to provide your manuscript number and title). Alternatively, you should make your own arrangements to improve the language quality and provide details in your response letter. – PeerJ Staff

Reviewer 2 ·

Basic reporting

Still there are so many english language usage errors in even in this revised version.

Experimental design

Authors have conducted limited review on existing literature. Experimental design is inadequate.

Validity of the findings

Authors have not compared their results with the recent existing works. Now in this revised version only one recent work has been added by the authors. It is suggested to compare their results with the existing works of related standard datasets.

Additional comments

Authors have not used the standard dataset in their work. Proposed methodology should be improved with still more clarity.

---

## Round 0.5 · accepted · Accept

Dear Authors,

One of the reviewers did not respond to the invitation to review the revised manuscript. One reviewer accepted the invitation but did not send his/her review. Due to time constraints, I have evaluated the revision myself and believe that your manuscript appears improved and ready for publication after the latest revision.

Best wishes,